# The Association between Anatomical Variants of Musculoskeletal Structures and Nerve Compressions of the Lower Limb: A Systematic Review and Meta-Analysis

**DOI:** 10.3390/diagnostics14070695

**Published:** 2024-03-26

**Authors:** Juan José Valenzuela-Fuenzalida, Alfredo Inostroza-Wegner, Francisca Osorio-Muñoz, Daniel Milos-Brandenberg, Andres Santana-Machuca, Pablo Nova Baeza, Mathias Orellana Donoso, Alejandro Bruna-Mejias, Joe Iwanaga, Juan Sanchis-Gimeno, Hector Gutierrez-Espinoza

**Affiliations:** 1Department of Morphology and Function, Faculty of Health Sciences, Universidad De Las Américas, Santiago 7500000, Chile; juan.kine.2015@gmail.com (J.J.V.-F.); andres.santana@gmail.com (A.S.-M.); 2Departament de Morfología, Facultad de Medicina, Universidad Andrés Bello, Santiago 8320000, Chile; alf.in.weg@gmail.com (A.I.-W.); panchi.osorio.m@gmail.com (F.O.-M.); pablo.nova@usach.cl (P.N.B.); mathor94@gmail.com (M.O.D.); abrunam@unab.cl (A.B.-M.); 3Escuela de Medicina, Facultad Ciencias de la Salud, Universidad del Alba, Santiago 8320000, Chile; danielmilos.b@gmail.com; 4Escuela de Medicina, Universidad Finis Terrae, Santiago 7501015, Chile; 5Department of Oral and Maxillofacial Anatomy, Graduate School of Medical and Dental Sciences, Tokyo Medical and Dental University, Tokyo 113-8510, Japan; iwanagajoeca@gmail.com; 6Department of Neurosurgery, Tulane Center for Clinical Neurosciences, Tulane University School of Medicine, New Orleans, LA 70112, USA; 7Department of Neurology, Tulane Center for Clinical Neurosciences, Tulane University School of Medicine, New Orleans, LA 70112, USA; 8Department of Structural & Cellular Biology, Tulane University School of Medicine, New Orleans, LA 70112, USA; 9GIAVAL Research Group, Department of Anatomy and Human Embryology, Faculty of Medicine, University of Valencia, 46001 Valencia, Spain; juan.sanchis@uv.es; 10Escuela de Fisioterapia, Universidad de las Americas, Quito 170137, Ecuador

**Keywords:** lower extremity, nerve compressions, sciatic nerve, femoral cutaneous nerve, fibular nerve, neural entrapment, anatomical variants, clinical anatomy

## Abstract

**Objective:** The aim of this study was to describe the main anatomical variants and morphofunctional alterations in the lower limb that compress surrounding nervous structures in the gluteal region, thigh region, and leg and foot region. **Methods:** We searched the Medline, Scopus, Web of Science, Google Scholar, CINAHL, and LILACS databases from their inception up to October 2023. An assurance tool for anatomical studies (AQUA) was used to evaluate methodological quality, and the Joanna Briggs Institute assessment tool for case reports was also used. Forest plots were generated to assess the prevalence of variants of the gluteal region, thigh, and leg. **Results:** According to the forest plot of the gluteal region, the prevalence was 0.18 (0.14–0.23), with a heterogeneity of 93.52%. For the thigh region, the forest plot presented a prevalence of 0.10 (0.03–0.17) and a heterogeneity of 91.18%. The forest plot of the leg region was based on seven studies, which presented a prevalence of 0.01 (0.01–0.01) and a heterogeneity of 96.18%. **Conclusions:** This review and meta-analysis showed that, in studies that analyzed nerve compressions, the prevalence was low in the thigh and leg regions, while in the gluteal region, it was slightly higher. This is mainly due to the PM region and its different variants. We believe that it is important to analyze all the variant regions defined in this study and that surgeons treating the lower limb should be attentive to these possible scenarios so that they can anticipate possible surgical situations and thus avoid surgical complications.

## 1. Introduction

Sir Herbert Seddon first described three levels of nerve injury in 1943: neuropraxia, axonotmesis, and neurotmesis. Neuropraxia involves a local conduction block in a nerve at the site of injury, temporarily interrupting nerve impulses. This injury is typically caused by compression. The analysis of nerve compression in corpses requires a morphometric study of the corpses to establish the relationship between the decreased space and the size of the nerve and to be able to infer a possible site of nerve compression [1]. One of the main factors affecting the functionality of the lower limb is entrapment neuropathies. These neuropathies result from the compression of a peripheral nerve, generally in typical sites where the nerve is vulnerable or from an abnormal occupation of the space along the nerve channels [2]. The main symptoms are pain, paresthesia, and muscle weakness in the area, among others [3].

Regarding the anatomy of the lower limb, the main nerves that contribute to the innervation of the region are the sciatic (SN), lateral femoral cutaneous (LFCN), femoral (FN), peroneal, saphenous, and tibial (TN) nerves [4]. The most common site of SN entrapment corresponds to compression by the PM when the nerve exits the pelvic cavity through the greater sciatic foramen, in most cases, under the muscle [4]. This neuropathy is known as PM. On the other hand, nerve compressions associated with the FN result from compression either in the retroperitoneal space or near the inguinal ligament [2]. As for the LFCN, it enters the thigh passing behind the inguinal ligament, just medial to the anterior superior iliac spine. The latter corresponds to the most common site of compression, causing the nerve compression syndrome known as meralgia paresthetica [4]. The common peroneal nerve (CFN) diverges from the SN over the popliteal fossa and descends laterally toward the leg; at this point, the CFN is closely associated with the periosteum of the fibula at its proximal epiphysis [4]. The common peroneal nerve (CFN) diverges from the SN over the popliteal fossa and descends laterally toward the leg; at this point, the CFN is closely associated with the periosteum of the fibula at its proximal epiphysis [4]. The region of the proximal epiphysis of the fibula is where damage occurs most frequently due to compression or traction of the CFN [5]. Regarding the saphenous nerve, compression is rare, but in some cases, it can occur just at the exit of the adductor canal, medially at the level of the knee [2].

Finally, the most common site of posterior tibial nerve compression neuropathy is the tarsal tunnel, where it typically occurs adjacent to muscle tendons and vascular structures [3]. This compression neuropathy is known as tarsal tunnel syndrome. Although there are typical sites in the lower extremities where nerves are more prone to compression, multiple studies have reported anatomical variants that pose a potential risk for nerve entrapment due to decreased space or other symptoms [6,7,8,9,10,11].

The objective of this study was to discover the main anatomical variants and morphofunctional alterations in the lower limb that compress its various nerve structures.

## 2. Methods

### 2.1. Protocol and Registration

This systematic review and meta-analysis were performed and reported according to the Preferred Reporting Items for Systematic Reviews and Meta-Analyses (PRISMA) statement [12] (Appendix A). This review has been registered in PROSPERO with the registration number CRD42022224166.

### 2.2. Eligibility Criteria

Studies on nerve compression in the limb associated with anatomical variants or anomalies of the surrounding structures in different regions of the lower limb were considered eligible for inclusion if they met the following criteria: (1) population: a sample of cadaveric dissections was analyzed through morphometry of the compromised region or images of patients with nerve compression associated with variants or anomalies of the lower limbs; (2) results: the presence of nerve compression in the lower limb was associated with the presence of anatomical variants of neighboring structures based on a morphometric analysis of cadavers and the analysis of a space reduction in the region through morphometry and imaging examinations in living patients with radiological findings of compression or symptoms thereof; (3) studies: this systematic review included research articles, research reports, or original research published in the English or Spanish language in peer-reviewed journals and indexed in some of the databases reviewed. The exclusion criteria were the following: (1) animal studies; (2) studies that analyzed only nerve compression without anatomical variants attributable to compression; (3) letters to the editor or comments.

### 2.3. Electronic Search

We systematically searched MEDLINE (via PubMed), Web of Science, Google Scholar, the Cumulative Index to Nursing and Allied Health Literature (CINAHL), and Scopus from their inception until April 2023. The search strategy included a combination of the following terms: “Entrapment nerve” (No Mesh), “variations anatomical” (No Mesh), “lower limb” (Mesh terms), “compression nerve” (No Mesh), “compression syndrome” (No Mesh), “clinical anatomy” (No Mesh), and “nerves low extremity” (No Mesh) using the Boolean connectors “AND”, “OR”, and “NOT”. The search strategies for the Medline database are available at the following link:

https://pubmed.ncbi.nlm.nih.gov/?term=%28%28%28%28Lower+extremity%5BTitle%2FAbstract%5D%29+AND+%28variations+anatomical%5BTitle%2FAbstract%5D%29%29+AND+%28compression+nerve%5BTitle%2FAbstract%5D%29%29+OR+%28compression+femoral+nerve%5BTitle%2FAbstract%5D%29%29+OR+%28compression+sciatic+nerve%5BTitle%2FAbstract%5D%29&filter=hum_ani.humans (accessed on 10 January 2024).

### 2.4. Study Selection

Two authors (JJV and MO) independently screened the titles and abstracts of references retrieved from the searches. Full texts were obtained for any references that either author considered to be potentially relevant. A third reviewer (DM) was consulted if a consensus could not be reached.

### 2.5. Data Collection Process

Two authors (AS and EL) independently extracted data on the outcomes of each study. The following data were extracted from the original reports: (i) authors and year of publication, (ii) country, (iii) type of study, (iv) sample characteristics (sample size, age, distribution, and sex), (v) prevalence and morphological characteristics of entrapments in the lower limb, (vi) statistical data reported by each study, (vii) main results.

### 2.6. Assessment of Methodological Quality of Included Studies

For the bias analysis of the included articles, we used a tool (AQUA) for the analysis of the methodological quality of anatomical studies [13]. As a measure of agreement, the agreement rate between reviewers was used using the kappa statistic; the reviewers were JJV, JM, and HGE. Two authors (JS and MR) separately assessed the risk of case study bias. To minimize the risk of bias, the case studies were analyzed by two authors (JJV and AB) using the Joanna Briggs Institute assessment tool for case reports [14]. This assessment tool has eight items, with answer options such as “yes”, “unclear”, “no”, or “not applicable”, to evaluate studies based on the following criteria: (1) low risk of bias: more than a 70% score of “yes”, (2) moderate risk of bias: a 50–69% score of “yes”, and (3) high risk of bias: less than a 49% score of “yes”. For publication bias, a funnel plot analysis was performed for the variables, and for bias in subgroups, a meta-regression analysis was conducted for the comparison of variants in the gluteal region.

### 2.7. Statistical Methods

The data extracted from the meta-analysis were interpreted through the calculation of the prevalence of lower limb variants using JAMOVI software (Version 2.0), BETA version [15]. For lower limb variants, the prevalence was calculated using the DerSimonian–Laird model with a Freeman–Tukey double-arcsine transformation to combine the summary data. Heterogeneity was evaluated with the heterogeneity statistic (I^2^). For the chi^2^ test, the *p* value, as proposed by the Cochrane Collaboration, was considered significant when it was <0.10. The values of the I^2^ statistic were interpreted with a 95% confidence interval [CI] as follows: 0–40% could reveal low heterogeneity; between 41% and 70% could indicate moderate heterogeneity; 71–100% could represent high heterogeneity.

## 3. Results

When analyzing the databases included in this review, we found 331 studies that met the search criteria using the keywords. The first criterion to eliminate results was duplicate articles, and the second was those that did not meet all the inclusion criteria or that met a previously defined exclusion criterion, which left a total of 175 articles for the analysis of full texts. After this, 97 that did not meet the criteria for data extraction were excluded (Figure 1). This resulted in 78 studies that met our inclusion criteria [10,16,17,18,19,20,21,22,23,24,25,26,27,28,29,30,31,32,33,34,35,36,37,38,39,40,41,42,43,44,45,46,47,48,49,50,51,52,53,54,55,56,57,58,59,60,61,62,63,64,65,66,67,68,69,70,71,72,73,74,75,76,77,78,79,80,81,82,83,84,85,86,87,88,89,90,91,92]. Since the lower limb is a structure with several regions and possible compression sites, the above is detailed in Table 1, Table 2 and Table 3.

### 3.1. Gluteal Region Variants

For the gluteal region, 29 studies were included [10,16,17,18,19,20,21,22,23,24,25,26,27,28,29,30,31,32,33,34,35,36,37,38,39,40,41,42,43], which analyzed only compressions of the SN, where the main compression site was in the region of the nerve passage in relation to the piriformis muscle (PM), which is the classic presentation of this nerve. The variants that were considered are those in which the SN runs below the PM or in which the SN passes through the PM by traversing or perforating its muscular belly. It is worth mentioning that, although there are more nerves in the gluteal region, such as the inferior gluteal nerve and superior gluteal nerve, in this study, the search strategy did not produce any studies that presented an association between the variant and symptomatic or asymptomatic compression.

### 3.2. Thigh Region Variants

Twenty-five studies were included for the thigh region [44,45,46,47,48,49,50,51,52,53,54,55,56,57,58,59,60,61,62,63,64,65,66,67,68]. These studies analyzed compressions of the following nerves: LFCN, FN, and ON. Twelve studies [44,45,46,47,48,49,50,51,58] analyzed compression of the LFCN, of which different typical compression sites were identified. The first site corresponds to the anterior superior iliac spine, where compression is either due to a change in the position of the tendons of the tensor fascia latae muscle and the tendinous origin of the sartorius muscle or due to a more lateral position of the muscle. Something similar occurs with the LFCN, which brings it closer to the muscles with insertion in the anterior superior iliac spine, increasing the predisposition to compressions. On the other hand, another compression site was due to a change in the position of the inguinal ligament and its proximity to the iliacus muscle, causing the inguinal ligament to be closer to the muscle, the contraction of which could gradually compress the iliac muscle in the LFCN. Another point of compression is due to a variation in which the nerve leaves the pelvic cavity through a bony canaliculus that perforates the iliac bone in its inferolateral region, called the Carai, and it can also be compressed by perforating the inguinal ligament. Finally, the proximal division of the femoral cutaneous nerve could generate compression under the inguinal ligament. Seven studies [52,53,54,55,56,57,60,63] analyzed the compression of the FN, which was compressed in the following regions: entrapment by the accessory belly of the iliacus muscle occurs where the NF divides into two branches, one that goes over the accessory iliacus muscle and another that passes between the iliacus muscle and the accessory belly, the latter being where the nerve will have possible compression; another variation that can generate compression of the NF is where the iliacus muscle covers the NF, or where the nerve pierces the iliacus muscle; and finally, there is a critical fibromuscular ring between the inguinal ligament and the lateral part of the iliopsoas muscle, but some accessory bellies of the greater psoas muscle can compress the FN. For the compression of the ON, four studies were included [65,66,67,68], one in which compression was produced by a surgical complication due to vaginal tape for the treatment of urinary incontinence, which intra-surgically produced an iatrogenic compression of the ON, leaving the patient with symptoms in the nerve innervation territory. Another variant associated with the compression of the ON is the presence of a supernumerary external obturator muscle, which causes the nerve to become trapped as it exits the pelvic region. Another study showed the entrapment of the ON associated with an endometriosis nodule. Lastly, another study showed intra-surgical complications in a patient with cervical surgery, which, during the procedure, generated the entrapment of the ON (Figure 2).

### 3.3. Leg Region Variants

For the region of the leg and foot, 21 studies were included [69,70,71,72,73,74,75,76,77,78,79,80,81,82,83,84,85,86,87,88,89], which analyzed compressions of the tibial, deep fibular, and superficial fibular nerves in both their common and terminal branches. Regarding the tibial nerve, the main compression associated with an anatomical variant was due to the presence of the accessory flexor digitorum longus muscle (FDAL), which proved to have a high variability among individuals who presented it. Compression occurred especially in those cases in which the muscle entered in the form of a muscle belly under the flexor retinaculum, although compression by the tendon of the same muscle has also been described. One study showed the presence of another supernumerary muscle, the medial perocalcaneal muscle, which, in conjunction with ALDF, also caused tarsal tunnel syndrome. Regarding the common fibular nerve, there are three points of probable compression related to anatomical variants: the deep muscular band to the superficial head of the long fibular muscle, the superficial muscular band to the deep head of the long fibular muscle, and the confluence of the origin of the soleus muscles and long fibular. The deep and superficial fasciae of the superficial head of the fibularis longus have also been associated with compression of the common fibular nerve [91]. Another variant related to the biceps femoris muscle, in which its short head is displaced distally and posteriorly, has been related to the compression of the common fibularis [92,93]. Another study indicated that the variant in which the superficial fibular nerve runs through the anterior compartment of the leg, instead of the lateral one, could be associated with a greater probability of generating superficial fibular entrapment (Figure 3).

### 3.4. Clinical Considerations

The clinical considerations differ regarding the anatomical variations of the SN. Lumbar or lumbosacral radiculopathy, also referred to as sciatica, refers to the compression of a nerve root at its exit from the intervertebral foramen or in the surrounding area. This condition was reported in two studies [30,31]. The first study had only 1 patient, while the second study analyzed 43 corpses. In the first study, the clinical condition was reported in the patient, whereas in the second, this condition was inferred based on the pathomechanical characteristics of the corpses, but without information about the possible symptoms presented by the patients. Piriformis syndrome is a clinical condition that occurs due to compression or impingement of the SN resulting from hypertrophy or contracture of the PM, and it can be further influenced by anatomical variations in it. This syndrome constitutes a set of signs and symptoms that are characterized by sensory, motor, and trophic alterations related to innervation by the SN. Only one study showed a relationship between the variations of the SN and PM syndrome [25], reporting it as a block of the SN, with a prevalence of 8%. This finding was a pathomechanical inference from the authors, as it was a cadaveric study involving 25 cadavers, and there were no reports of clinical symptoms. Ischialgia is clinically defined as pain that radiates (hypoesthesia-paresthesia or dysesthesia) along the course of the SN roots, and it is one of the manifestations of lumbar pain due to the impingement of the SN. Only one study showed a possible ischialgia clinical report. The foregoing was a pathomechanical inference from the authors since it was a cadaveric study, and this clinical condition was not technically analyzed. In the study, 65 cadavers were evaluated [35], showing a prevalence of variations of the SN of 21.54%. Variations of the SN not only contribute to the generation of pathologies but also, as reported in several studies, are crucial to understand in order to avoid complications in surgical procedures of the pelvic region, as well as the upper and posterior third of the thigh. Finally, four studies included in this review did not report a clinical correlation with variations of the SN [23,25,32,34]. Regarding the thigh region, there are different clinical presentations and distinctive syndromes depending on the nerve that is compressed within the area. The specific symptoms vary depending on the etiological cause and the degree of nerve involvement. Meralgia paresthetica is a clinical condition characterized by pain, numbness, paresthesia, and lateral thigh pain that occurs due to compression of the lateral cutaneous femoral nerve [93]. This condition does not imply muscle weakness, since the innervation of the nerve only causes sensitivity. FN compression has a broader clinical presentation, as it compromises skin sensation and certain motor functions. Its symptoms include pain, numbness, hypoesthesia, and paresthesia in the frontal aspect of the thigh and weakness or atrophy of the quadriceps muscle group, and, in some cases, there may be involvement of the patellar reflex. In addition, symptoms may occur in the anteromedial region of the knee, in the medial thigh, or even in the foot when the saphenous branch is compromised [94]. Obturator neuropathy is characterized by pain, numbness, paresthesia, and hypoesthesia in the medial thigh or groin region, which can easily be confused with conditions affecting the hip or inguinal region [95]. In certain cases, sensory compromise can extend into the knee joint and even down the leg. Obturator dysfunction will also present compromised motor function, since it innervates the adductor muscles of the thigh: the adductor longus, adductor brevis, and gracilis [96]. Clinical considerations related to anatomical variants in the leg region vary depending on the structures involved. The presence of an accessory soleus muscle was reported in a study [88], which presented four patients with the variant, three of them symptomatic and one asymptomatic. Symptomatic patients manifested chronic pain in the ankle area, which, in some cases, was accompanied by inflammation. In the physical examination, it was possible to identify atrophy at the calf level and tenderness in the region of the Achilles tendon and the tendon of the posterior tibial muscle. Moreover, one patient presented subluxation of the tendons of the peroneal muscles. The presence of an accessory flexor digitorum longus muscle was described in 18 studies of different characteristics [71,72,73,74,75,76,77,78,79,80,81,82,83,84,85,86,87,88]. In most cases, its presence was associated with tarsal tunnel syndrome, which is triggered by the compression of the tibial nerve within the tarsal tunnel, as this accessory muscle compresses the nerve. Patients with symptoms of tarsal tunnel syndrome manifested severe and chronic pain in the ankle region and sometimes in the heel region [75,82], which was accompanied by inflammation in the area in certain patients. In some cases, pain radiated along the posterior tibial nerve in the lower leg [83]. Some patients manifested paresthesia [76]. It is important to consider that there were rare cases of asymptomatic patients who presented the muscular variant [74,89].

### 3.5. Characteristics of the Study Sample

For the 28 studies of the gluteal region, the demographic characteristics are as follows: 11 studies were from the Asian continent, 8 were from Europe, 5 were from North America, 4 were from Africa, and 1 was from South America. Regarding the gender of the sample, only 19 studies reported whether the sample was male or female, among which 550 were male and 661 were female, which is equivalent to 45.4% and 54.6%, respectively, of the total sample of studies of the gluteal region. Regarding the demographic distribution among the 29 studies included for the thigh region, 15 studies were from Europe, 5 studies were from North America, and 8 studies were from Asia. Regarding the gender of the samples included, only 23 studies reported the gender of the sample, among which 160 of the total sample comprised females (42.4%), and 217 of the total sample comprised males (57.6%). Among the 21 studies included for the leg region, 2 studies were from Europe, 12 studies were from North America, and 5 were from Asia. In total, 14 studies reported the gender of the sample, among which 437 of the reported sample were female sex (36.4%) and 763 were male (63.6%).

### 3.6. Prevalence of Anatomical Variants Associated with Nerve Compression and Risk of Bias

Three forest diagrams were made to visualize the prevalence of anatomical variants associated with nerve compression of the lower limb. It should be noted that the prevalences shown in the diagrams only reflect the populations in the studies investigating the presence of variants in the lower limb associated with nerve compression and do not reflect either a global or regional population. For the forest plot of the gluteal region, 26 studies [16,17,18,19,20,21,22,23,24,25,26,27,28,29,30,31,32,33,34,35,36,37,38,39,40,41] presented a prevalence of 0.18 (0.14–0.23) and a heterogeneity of 93.52% (Figure 4). For this comparison, the funnel plot graph showed symmetry, indicating the absence of publication bias or factors influencing the results (Figure 5). We performed a meta-regression for the presence of lower limb nerve compressions associated with anatomical variants, which shows whether the data present a low bias and whether they can be included in a meta-analysis based on the dependent and independent variables (Figure 6). The thigh region prevalence forest plot was based on five studies [52,55,59,60,63], which presented a prevalence of 0.10 (0.03–0.17) and a heterogeneity of 91.18% (Figure 7). For this comparison, the funnel plot graph showed asymmetry, indicating the presence of publication bias or factors influencing the results (Figure 8). For the forest plot of the leg region, seven studies were included [70,71,72,73,75,79,81]. The prevalence was 0.01 (0.01–0.01) with a heterogeneity of 96.18% (Figure 9). For this comparison, the funnel plot graph showed asymmetry, indicating the presence of publication bias or factors influencing the results (Figure 10). For the bias of the studies included with AQUA, in the domain of the objectives and characteristics of the included studies, a low risk of bias was presented; in the domain of the study design, a low risk of bias was presented. The domain of methodological characteristics obtained a low risk of bias; the anatomical description domain showed low bias; and finally, the reporting of the outcome domain also presented mostly a low risk of bias. For the case studies, 29 studies were included, and only 2 presented a moderate level of risk of bias [77,79] (Figure 11 and Figure 12).

## 4. Discussion

This systematic review and meta-analysis aimed to report the prevalence of musculoskeletal variants or alterations in the lower limb that could cause some type of nerve entrapment, as well as define specific points of entrapment in order to provide diagnostic guidelines and an improved therapeutic approach. A calculation of the prevalence of the studies that met the inclusion criteria was made to show that, in addition to the classic points of compression, which are not the only factors that are associated with functional and postural components, on many occasions, the cause of the compression is a change in the morphology of the nerve, its aberrant path, or some structure coming in close proximity to the path that can produce compression syndrome. The main result found in our review is that a significant number of variants exist that cause nerve compression and that knowing them will improve the accuracy of diagnosis and avoid a differential diagnosis bias.

Compared to other articles that have shown the relationship between anatomical and clinical nerve compressions of the lower limb and different surrounding musculoskeletal structures, our review presents a highly detailed anatomical and clinical approach to nerve compressions of the lower limb and clearly reports the structures that can generate compressions due to surrounding anatomical variants. Regarding other similar reviews, in our search, we did not find any reviews in the literature that made the association that we propose as an objective, but there are some that present a relationship with compressions, and we will detail them below. First, we will mention the 2021 Orellana review [95], which sought to compare compressions of the upper limb and dealt with the same relationship with anatomical variants. Another review is that by Craig (2013) [4], who showed how the nerves of the lower limb, when presenting a condition of entrapment, cause the segment to have a loss of functionality or sensitivity alterations. He focused his study on demonstrating the gold standard for the diagnosis of compression sites using electromyography. The Bowley 2019 review [3] showed that compressive neuropathies do not present symptoms such as coordination and muscle weakness, and with this, the paralysis of different nerves can be associated in a functional analysis. Finally, this review did not show any relationship with variants or abnormalities of the surrounding musculoskeletal structures. The review by Thoma (2003) [96] showed that the compression of a nerve of the lower limb is disabling. Although the study names terms or anatomical structures that could produce it, it does not detail what they could be specifically, as detailed in this review. Finally, the review by Madani (2020) [2] found an association between lower limb nerve compressions and rheumatological and systemic disorders, giving recurrent functional alterations in the lower limb as symptoms. However, similar to the aforementioned reviews, this study did not make an association with alterations of the surrounding structures. The included studies were of various methodological types, but those used to analyze the prevalence of anatomical variants and abnormalities associated with nerve compressions of the lower limb were retrospective and prospective observational studies, as well as case reports and case series. Regarding the characteristics of the sample, it was quite homogeneous between men and women for the three regions studied; therefore, we cannot infer whether the presence of nerve compressions due to anatomical variants or abnormalities has any association with the sex of the sample. Regarding the geographical distribution, as the distribution was quite heterogeneous, the most predominant being Europe, North America, and Asia, it cannot be inferred whether specific variants and compression are associated with certain races or ethnic groups. The prevalence of nerve compressions associated with anatomical variants of the limb was 18% for the gluteal region, 10% for the thigh region, and 9% for the leg region. We understand that these values may present a high level of bias due to the high heterogeneity of the samples, since some studies presented a high number of variants, and several authors stated that their intention was to study the variant directly without a random sample, which might mean that samples that did not present the variant were often excluded.

Regarding the anatomoclinical considerations of the structures studied in this review, in the gluteal region, the main nerve that is compressed and presents recurrent symptoms in the gluteal region and posterior aspect of the lower limb is the sciatic nerve. The included studies showed that this nerve is compressed mainly in the passage in relation to the PM. This muscle can sometimes compress the nerve due to changes in the path of the same nerve, but if we associate it with changes in a neighboring structure, changes in the normal morphology of the nerve occur in the PM, which can present divisions in its muscle belly or present an additional muscle belly that can compress the sciatic nerve without changing its path; as mentioned above, this is the main structure that can compress the sciatic nerve according to the included studies, but one study mentioned that changes in the sacrotuberous ligament could also cause proximal sciatic nerve compression. Regarding the clinic, most studies report that, although nerve compression should be symptomatic based on compression only with the PM, giving the symptoms a structural component, the main symptomatology is due to mechanical and not necessarily structural alterations. This is based on the fact that the largest population that presents with symptoms does not have any type of variation or structural change, the latter being attributed to the muscle mass ratio of the PM and the caliber of the nerve. In the thigh region, we found three nerves that could classically present compression: the lateral femoral cutaneous nerve, the femoral nerve, and the obturator nerve. Of these three, the lateral femoral cutaneous nerve has the highest prevalence of compression and clinical considerations from the point of view of changes in neighboring structures that can compress it. These changes are mainly in the proximal insertion of the sartorius muscle and tensor fascia latae muscle, which, being located more medially, cause compression of the lateral femoral cutaneous nerve. Another point of compression is by a variant in the position of the inguinal ligament, which, when it is more ascended, produces compression in the passage of the nerve. Finally, a canaliculus formed in the iliac bone called the Carai, where the lateral femoral cutaneous nerve crosses it, can compress the passage of the nerve. Of the three previously mentioned, the one that has been associated with the worst symptoms in the lateral region of the thigh is compression by the Carai, and the smaller its diameter, the less space it will have and the more susceptible it becomes to compression. For compressions of the femoral nerve associated with variants of the surrounding structures, the main variants were an accessory belly of the iliacus muscle, which caused the nerve to run between the iliacus muscle and this accessory belly, which, in the face of very demanding contractions, especially flexion of the hip, causes compression of the femoral nerve, generating symptoms in the anterior region of the thigh. Another structure is one in which the iliacus muscle covers the femoral nerve, although, here, there is no structural variant but rather a positional variant that compresses the femoral nerve. Finally, an accessory belly toward the medial side of the psoas major muscle could also generate compression of the femoral nerve due to space problems.

All of the above will have an almost identical clinical presentation: as shown, they are functional alterations of the hip flexor musculature and pain in the anterior region of the thigh. For the compression of the obturator nerve, one of the points was due to an alteration in an intra-surgical process due to vaginal tape, which is attributed more to a bad intra-surgical maneuver than something more structural, for which it is recommended in these surgeries to take great care with the obturator nerve. An associated variant is the supernumerary internal obturator muscle, which generates a decrease in the space of the sciatic nerve exit toward the medial region of the thigh, generating recurrent symptoms of the obturator nerve. Finally, a study showed the presence of an endometriotic nodule that produced asymptomatic compression of the obturator nerve, which was evidenced intra-surgically. Obturator nerve compressions have been little studied clinically, but they could be a fundamental part of many differential diagnoses. Finally, compressions of nerves in the leg region were mainly associated in the literature with compression of the tibial nerve, the deep fibular nerve, and the superficial fibular nerve. The compression of the tibial nerve due to a surrounding anatomical variant was mainly associated with the LADF, which compresses the tibial nerve in the tarsal tunnel, giving the same symptoms and being one of the possible differential diagnoses for tarsal tunnel syndrome. For compression of the superficial fibular nerve, the variant structure would be mainly a muscle accessory soleus or long fibular in its proximal origin with an internal belly, which could cause the nerve to be compressed proximally. Finally, the common fibular nerve could be compressed by a biceps femoris muscle with a displacement that compresses the nerve, which is only associated with a change in position. Although the compression points are varied, they should be known in detail, and which of them will be more predisposed to produce symptoms associated with their own pathomechanical factors should be understood. Given the presence of these symptoms and having ruled out other possible prevalent pathologies, it is important to perform imaging and electrophysiological studies that allow the observation of some nerve compressions. For this reason, some studies recommend that any detection of this anatomical variant be reported as a finding during a routine analysis so that the corresponding preventive measures can be taken and patients can be monitored, which minimizes the risk of symptoms.

### Limitations

This study had some limitations. First, a publication bias could exist among the included studies, since studies with different results that were found in the non-indexed literature in the selected databases may have been omitted. Another limitation is the possibility of not having carried out the most sensitive and specific search regarding the topic to be studied. Finally, the selection of articles may have been influenced by the personal preferences of the authors, although we tried to minimize such a bias as much as possible.

## 5. Conclusions

This review and meta-analysis demonstrated that the prevalence of anatomical variants in studies that analyzed nerve compressions was low in the thigh and leg regions, while in the gluteal region, it was slightly higher, this being mainly due to the region of the PM and its different variants. In relation to the above, we highlight that, in the presence of this association, compression syndromes of different nerves of the lower limb may occur. We also highlight that, among the main nerve compressions of the lower limbs, we were able to demonstrate through the literature that, for example, for meralgia paresthetica, which affects the LCFN, there are a significant number of variants of surrounding structures that produce nerve compression. Finally, based on this research, we believe that it would be beneficial to analyze all the variant regions defined in this study and for surgeons in charge of addressing the treatment of the lower limb to be aware of these possible scenarios and thus anticipate possible surgical complications.

### Investigation Perspectives

Nerve compressions are alterations in the lower limb that occur recurrently. In most cases, the diagnosis is attributed to a mechanical or postural factor that triggers the symptoms associated with the compression. With this review, we want to add that there are also some anatomical variants of surrounding structures that could generate compression of the nerve and also generate symptomatology in the region of innervation, which is why we believe that knowing these variants can help clinicians avoid differential diagnoses and, consequently, also generate timely and effective treatments for patients who present compression due to an anatomical variant.

## Figures and Tables

**Figure 1 diagnostics-14-00695-f001:**
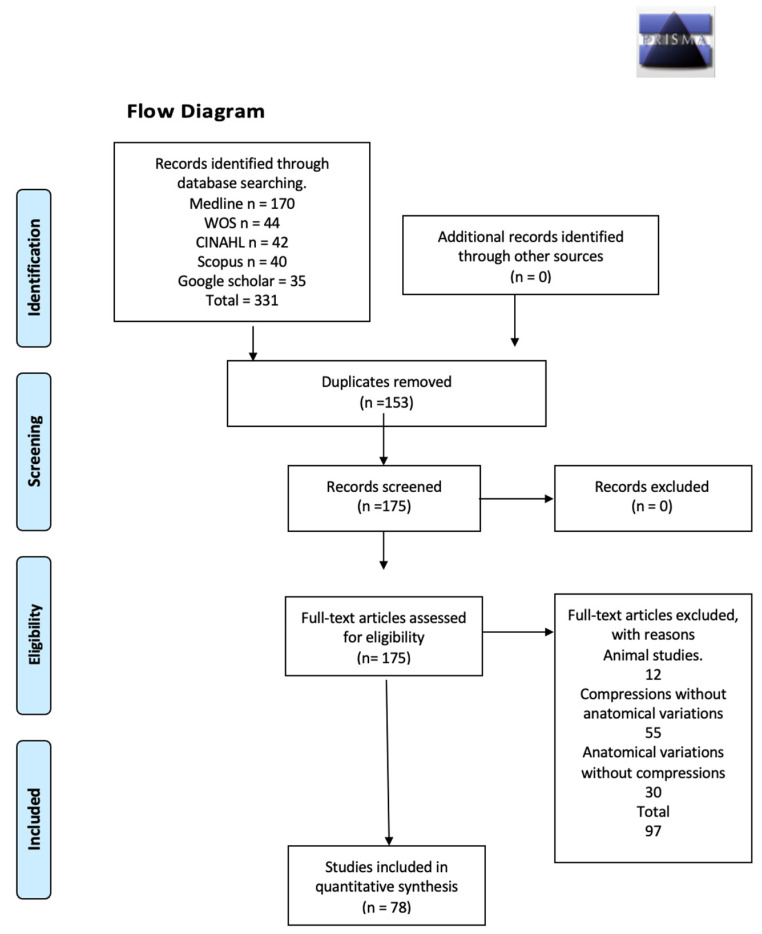
Flow chart.

**Figure 2 diagnostics-14-00695-f002:**
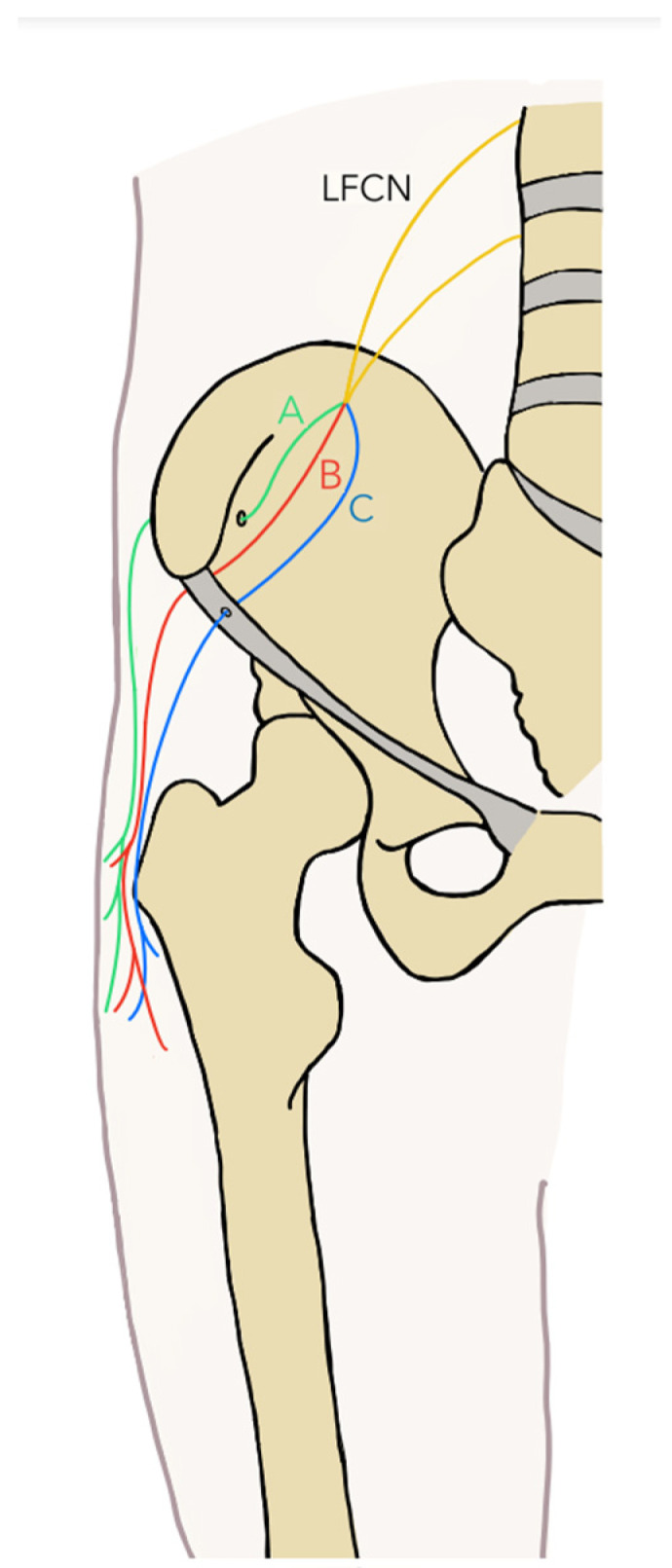
Compressions of thigh region.

**Figure 3 diagnostics-14-00695-f003:**
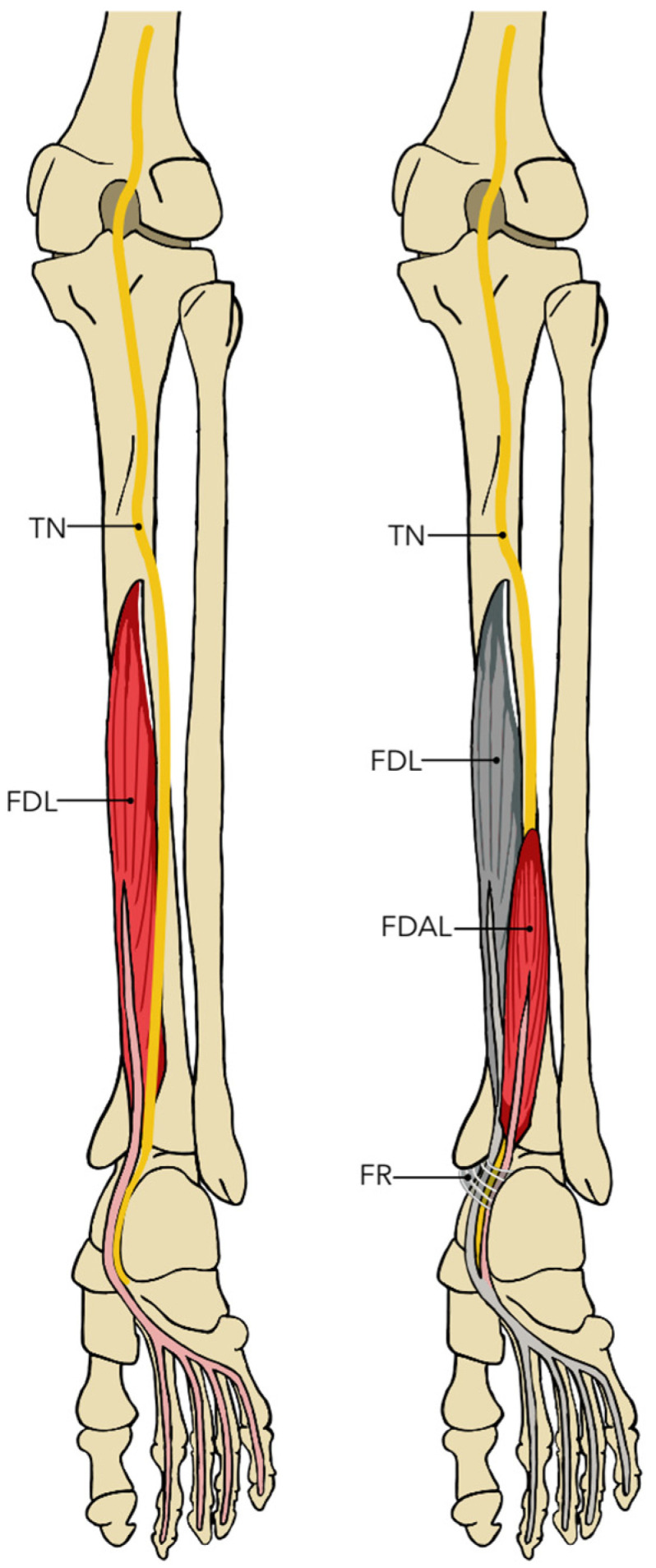
Compressions of leg region.

**Figure 4 diagnostics-14-00695-f004:**
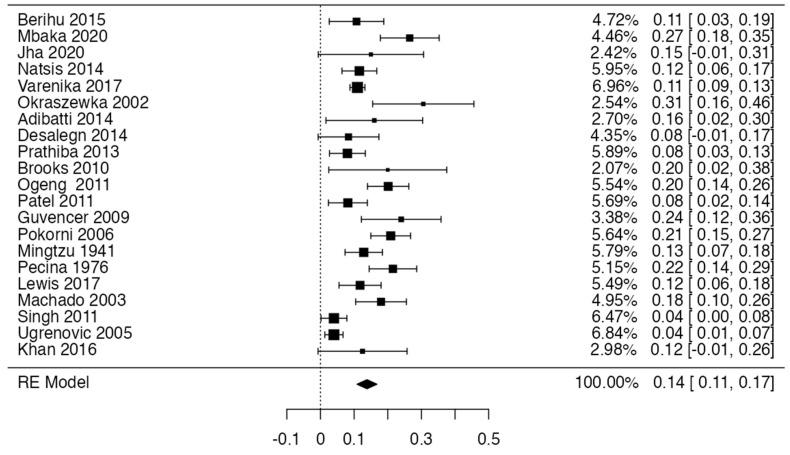
A forest plot for the prevalence of nerve compressions of the gluteal region associated with anatomical variants of surrounding structures [16,17,18,19,20,21,22,23,24,25,26,27,28,29,30,31,32,33,34,35,36,37,38,39,40,41].

**Figure 5 diagnostics-14-00695-f005:**
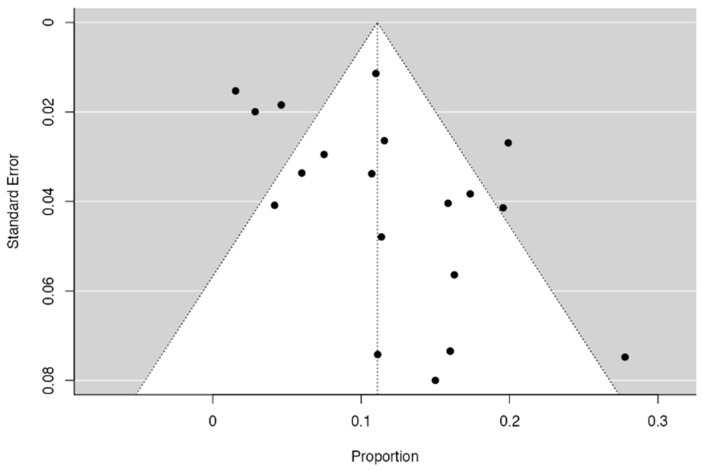
The funnel plot of nerve compressions of the gluteal region associated with anatomical variants of surrounding structures.

**Figure 6 diagnostics-14-00695-f006:**
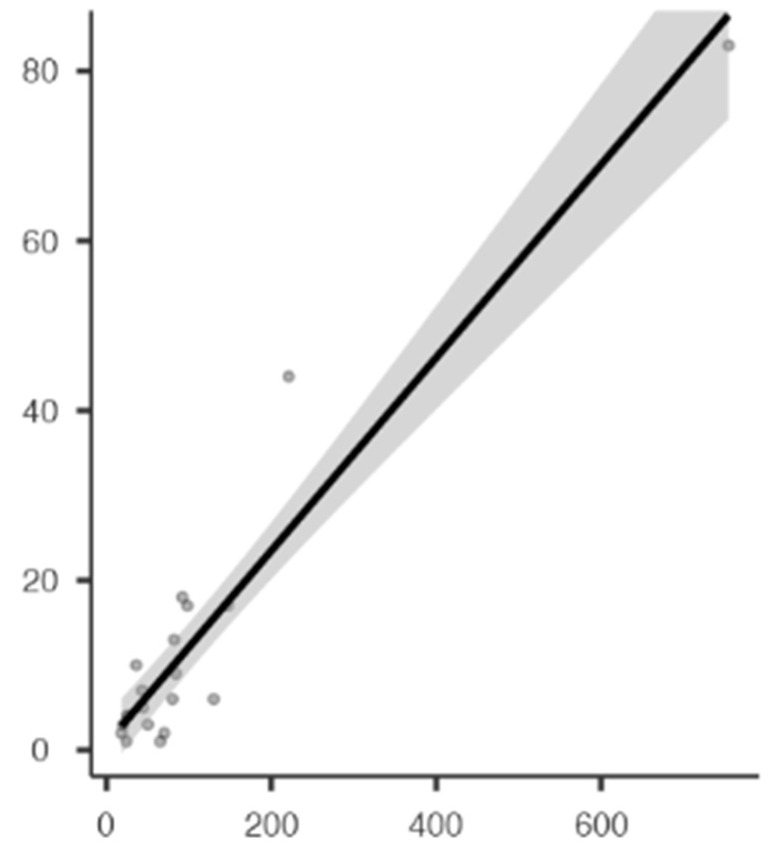
Meta-regression of compressions of the gluteal region associated with anatomical variants of surrounding structures.

**Figure 7 diagnostics-14-00695-f007:**
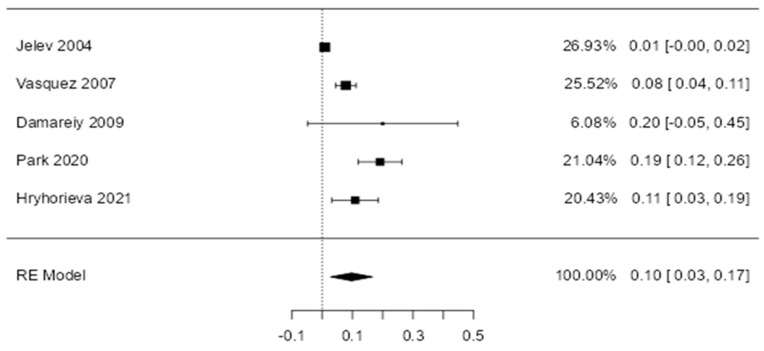
A forest plot for the prevalence of nerve compressions of the thigh region associated with anatomical variants of surrounding structures [53,55,59,60].

**Figure 8 diagnostics-14-00695-f008:**
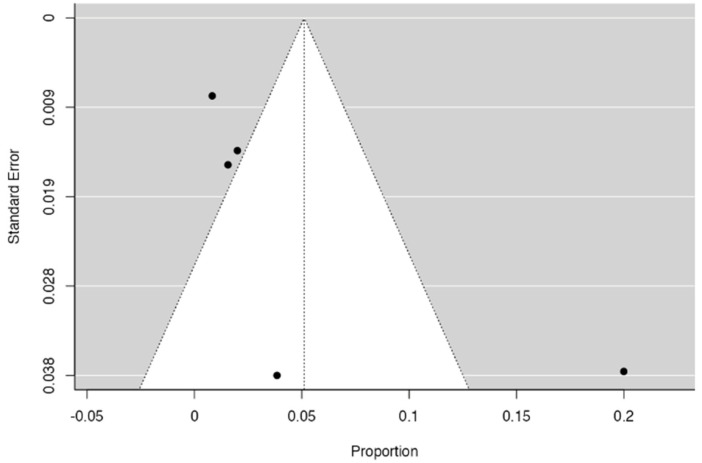
A funnel plot of nerve compressions of the thigh region associated with anatomical variants of surrounding structures [53,55,59,60].

**Figure 9 diagnostics-14-00695-f009:**
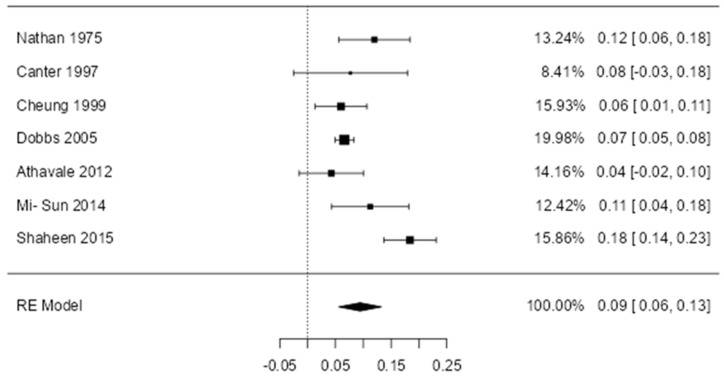
A forest plot for the prevalence of nerve compressions in the thigh region associated with anatomical variants of surrounding structures [70,72,73,75,79,81,86].

**Figure 10 diagnostics-14-00695-f010:**
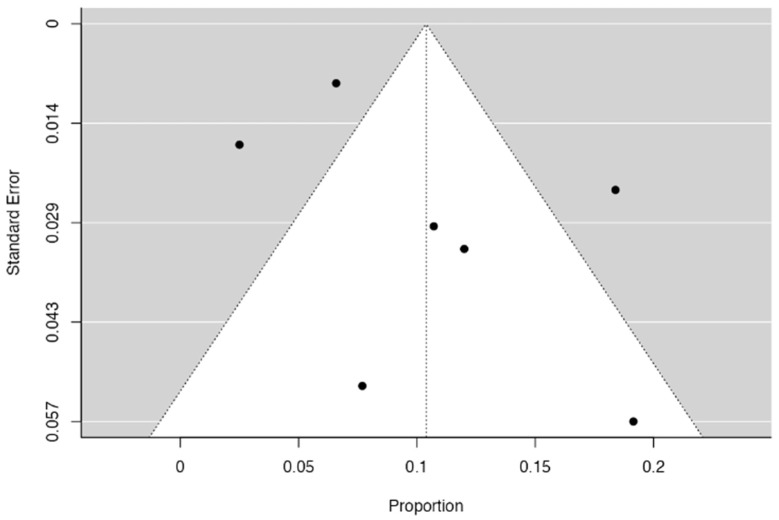
A funnel plot of nerve compressions in the thigh region associated with anatomical variants of surrounding structures [70,72,73,75,79,81,86].

**Figure 11 diagnostics-14-00695-f011:**
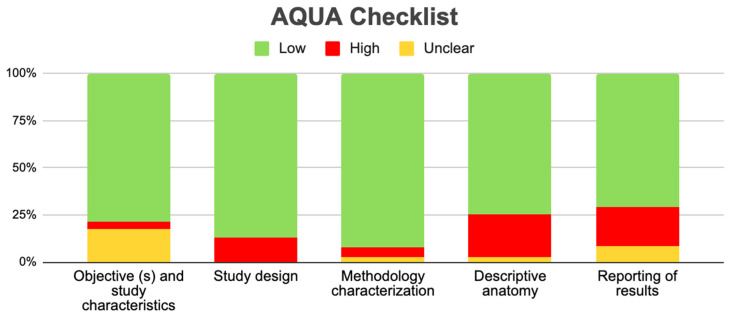
AQUA checklist graphic.

**Figure 12 diagnostics-14-00695-f012:**
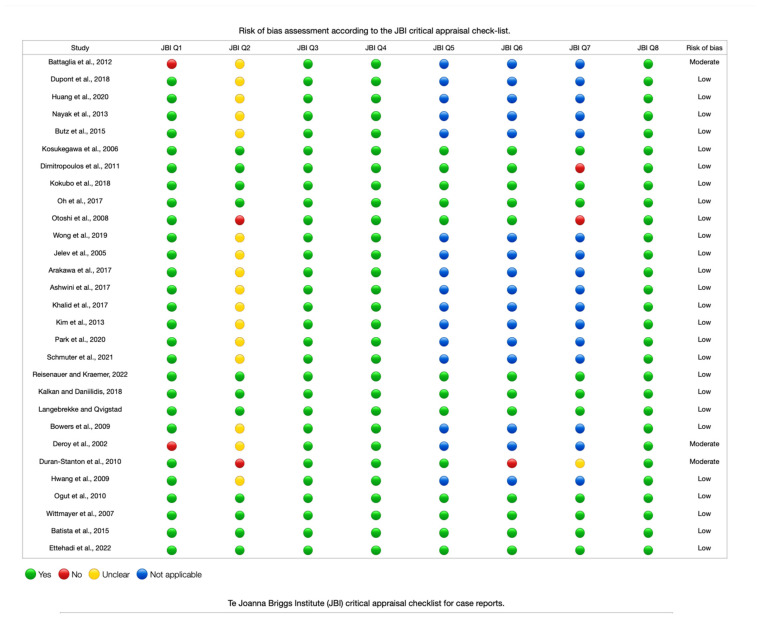
Risk of bias in case studies [77,78,79,80,81,82,83,84,85,86,87,88,89].

**Table 1 diagnostics-14-00695-t001:** Gluteal region.

Author and Year	Type of Study	Incidence and Characteristics	Nerve Entrapment	Statistical Values and Characteristics	Geographic Region	Gender	Laterality
Carai et al. (2009) [44]	Case study, 148	The course of the LFCN was evaluated in 148 patients operated on due to MP. A subfascial course was described in 131 cases and an epifascial one in 4 cases. In 115 cases, the nerve left the pelvis below the inguinal ligament; in 12 cases, it left through a cleft in the ligament; and in 6 cases, it exited through a bony canaliculus in the iliac bone. In 84 cases, the LFCN bifurcated normally, and in 51 cases, it presented early bifurcation.	Yes, LFCN	Not included	Italy	Not specified	7 bilateral, 141 unilateral cases
Dimitropoulos et al. (2011) [45]	Case study, 1	The patient manifested pain and paresthesia in the upper lateral area of the thigh due to MP. During surgery, an LFCN variation was reported: instead of crossing below the inguinal ligament, it crossed over the iliac crest.	Yes, LFCN	Not included	Switzerland	Female	Left
Kokubo et al. (2018) [46]	Case study, 1	The patient presented with pain and paresthesia in the anterolateral region of the thigh. During surgery to decompress the LFCN, it was identified that the nerve penetrated the inguinal ligament together with a thick NF and then formed a curve and became trapped in connective tissue.	Yes, LFCN	Not included	Japan	Male	Right
Erbil et al. (2002) [47]	Cadaveric study, 28	A variation of the LFCN was observed in two cases. In case 1, 2 LFCNs were present: a lateral nerve that ran under the inguinal ligament and a medial nerve that passed anterior to the iliacus muscle over the inguinal ligament. In case 2, there were 3 LFCNs that perforated the psoas major muscle; the superior and inferior nerves passed under the LI, and the middle nerve passed through the LI.	Not specified	Not included	Turkey	22 males, 6 females	Not specified
Tomaszewski et al. (2016) [46]	Case study	A meta-analysis of 24 studies was performed to determine the prevalence of LFCN variations. Seven types of pelvic outlets were identified, with 5 branching patterns. It was found that the grouped mean distance from the exit point of the LFCN to the ASIS was 1.90 cm.	Not specified	The oddest pattern of exit is through the pelvis, over the IL, with a prevalence of 0.9%. The bifurcation inside the pelvis has a prevalence of 11.8%.	Poland	Not specified	Not specified
De Ruiter et al. (2021) [47]	Systematic review	The course of the LFCN was determined by ultrasound prior to MP correction surgeries. Five types of routes (A, B, C, D, and E) were established for the inguinal ligament leaving the pelvic cavity.	Yes, LFCN	Of 54 patients with idiopathic MP, 79% are type B, 9% type C, 5% type D, 7% type E and 0% type A.	The Netherlands	Not specified	Not specified
Chhabra et al. (2013) [48]		Two independent readers reviewed MR for the evaluation of MP. Signal intensity was measured at 3 points along the course of the LFCN: proximal, distal, and at the ASIS level. The reliability of the use of MRI for PM diagnoses was demonstrated.	Not specified	The specificity and sensitivity of both readers were >71% and >94 respectively.	USA	Not specified	Not specified
Oh et al. (2017) [48]	Case report, 1	A patient aged 64 presented with paresthesia on the lateral aspect of both thighs after a femoral cannulation procedure. The confirmation of MP was made by a sensory nerve conduction study.	Yes, LFCN (procedural obliteration)	Not included	Republic of Korea	Male	Bilateral
de Ridder et al. (1999) [49]	Cadaveric study, 200	Variants of the LFCN course were studied in 200 cadavers. In addition, patients with LFCN complications after pelvic surgery were identified to determine the incidence and persistence of postoperative alterations.	Yes, LFCN (procedural obliteration)	51 cadavers (25.5%) presented an abnormal course of LFCN.	The Netherlands	Not specified	Not specified
Moritz et al. (2013) [50]	Case study, 28	The anatomical course and compression site of the LFCN were evaluated using high-definition ultrasound in 28 patients with idiopathic MP.	Yes, LFCN	In patients with MP, the mean distance between the LFCN and the ASIS was 0.52 cm; 1.27 cm less than in the control group	Austria	20 males, 8 females	Not specified
Otoshi et al. (2008) [51]	Case report, 1	The patient reported dysesthesia in the anterolateral thigh. The LFCN was compressed by the iliacus muscle against the inguinal ligament and sheathed at the tendinous origin of the sartorius. Neurolysis and partial dissection alleviated symptoms.	Yes, LFCN	Not included	Japan	Male	Left
Jelev et al. (2004) [52]	Cadaveric study, 100	Anatomical dissection revealed bilateral variations of the psoas major and iliacus muscles combined with variations of the left and right femoral nerves. The left accessory iliopsoas was formed by the connection of two accessory muscles: the accessory psoas major and the accessory iliacus.	Not specified	Not included	Bulgaria	Female	Bilateral

**Table 2 diagnostics-14-00695-t002:** Thigh.

Wong T a et al., 2019 [53]	Case study, 1	A variant of the psoas muscle called psoas quartus generates a multiple division of the femoral nerve. The medial branch descends posterior to the muscle, the second and third branches surround it and converge lateral to it, and the lateral branch descends anteroinferior to it.	Not specified	Not included	USA	Female	Left
Vázquez et al., 2007 [54]	Cadaveric study, 121	A cadaveric study with a sample size of 121 describes anatomical variations of the iliacus and psoas muscles that may be involved in the entrapment of the femoral nerve. The most prevalent variation was the piercing of the femoral nerve by a muscular sheet.	Not specified	*p* < 0.05	England	64 male, 57 female	Bilateral
Tiraj Parikh et al., 2019 [55]	Cadaveric study, 1	A case of a variant of the psoas muscle called psoas tertius is reported. It divides the femoral nerve into an anterior branch and a posterior branch. Both surround the psoas tertius muscle and then converge on the lateral border of the psoas major muscle. 1/60	Not specified	Not included	USA	Female	Left
Arakawa et al., 2017 [56]	Case study, 1	In the posterior thigh, there are four extra muscles. They originate from the greater trochanter (Muscle I), the long head of the biceps femoris (Muscles II and III), and the adductor magnus (Muscle IV). They fuse in various ways, forming additional connections between the biceps femoris and surrounding structures.	Not specified	Not included	Japan	Male	Right
Ashwini et al., 2017 [57]	Case study, 1	A femoral nerve is shown bifurcating after originating from the lumbar plexus. The superior division passes superficial to the iliacus muscle and the posterior division passes deep to the muscle. Both divisions converge just before crossing the inguinal ligament inferiorly.	Not specified	Not included	India	Male	Left
Damerey et al., 2008 [58]	Case study, 26	Anatomical variations of the LFCN course could be seen where the nerve exits the pelvis. The LFCN gave the femoral branch of the genitofemoral nerve before exiting the pelvis.	Not specified	Not included	France	5 unspecified, 13 males, and 8 females	Bilateral
Hryhorieva et al., 2021 [59]	Case study, 64 fetal lower limbs	Considering that the terminal branches of the adjacent cutaneous nerves of the femoral region intersect and overlap, innervation shunts are formed, due to which, in the case of possible damage to one of the nerves, the other will compensate to some extent for its insufficiency. Anastomoses were found between the cutaneous nerves in the form of loops of various shapes and sizes, namely, between the cutaneous-fascia branches of the femoral and ilioinguinal nerves and the femoral and obturator nerves.	Not specified	Not included	Ukraine	Not specified	Bilateral
Khalid et al., 2017 [60]	Case study, 1	A variant of the psoas muscle called psoas tertius is presented, which divides the femoral nerve into a thick lateral branch and a thin medial branch, which surround the muscle and then converge into a single femoral nerve.	Not specified	Not included	USA	Female	Right
Kim et al., 2014 [61]	Case study, 1	A muscular variant is located transversely in the superficial region of the popliteal fossa. It extends from the medial head of the gastrocnemius muscle to the tendon of the long head of the biceps femoris, covering the neurovascular structures.	Not specified	Not included	Republic of Korea	Male	Right
Park JA et al., 2020 [62]	Case study, 115	The CPN typically decreased posteriorly to the SHBFM and then superficially to the LGCM. However, about 20% of the time, the BFM (especially the SHBFM) extended more distally and posteriorly. The CPN crossed a tunnel formed between the SHBFM and the LGCM.	Not specified	Not included	Republic of Korea	68 male, 47 female	Bilateral
Park JH et al. (2020) [63]	Case study, 1	An accessory belly of the iliacus muscle is presented that divides the femoral nerve into a thin branch that runs superficial to the accessory muscle and a thick branch that runs between the accessory muscle and the iliacus muscle.	Yes, femoral nerve	Not included	Republic of Korea	Male	Left
Schmuter G et al., 2021 [64]	Case study, 1	A variant of the hamstring muscle consists of a fusion between the semitendinosus muscle and the long head of the biceps femoris muscle through a common belly that originates from the ischial tuberosity. A unilateral accessory muscle bundle is present that projects from the gluteus maximus muscle and fuses with the long head of the biceps femoris muscle.	Not specified	Not included	USA	Female	Right
Reisenauer and Kraemer, 2022 [65]	Case study, 1	A 52-year-old woman presented with right hip and groin pain and muscle weakness in the thigh after surgery. Right obturator entrapment was demonstrated by electromyography.	Not specified	Not included	Germany	Female	Right
Proudhon et al., 2023 [66]	Cadaveric study, 9	The endopelvic and exopelvic course of the obturator nerve and its relationship with the obturator canal and with the obturator, external obturator, and long, short, and greater adductor muscles were studied.	Not specified	In 44% of cases, the obturator nerve passes through the internal obturator muscle. In 11% of cases, it divides outside the obturator canal. In 22% of cases, the posterior branch passes through, and in 33% cases, the anterior branch attaches to the fascia.	France	8 male, 1 female	Not specified
Kalkan and Daniilidis, 2018 [67]	Case report, 1	A case of a deeply infiltrating endometriotic nodule that trapped the obturator nerve is reported. It manifested with chronic groin pain, which disappeared immediately after laparoscopic surgery.	Yes, obturator nerve	Not included	Turkey	Female	Right
Langebrekke and Qvigstad [68]	Case report, 1	Following pelvic lymphadenectomy and trachelectomy for cervical cancer, symptoms of obturator nerve entrapment were present. Endometriosis surrounding the nerve was discovered. Laparoscopic surgery was performed, and the symptoms disappeared.	Yes, obturator nerve	Not included	Norway	Female	Right

**Table 3 diagnostics-14-00695-t003:** Leg region.

Author and Year	Type of Study	Incidence and Characteristics	Entrapment Association	Statistical Values and Characteristics	Geographic Region	Gender	Laterality
Athavale et al. 2012 [79]	Cadaveric study, 47 lower limbs	The medial head of the accessory flexor digitorum muscle (quadratus plataris) extended through the tarsal tunnel in 80% of cases. The presence of the accessory muscle FDAL was observed in two cases.	Yes, tarsal tunnel syndrome	Not included	India	Not specified	Bilateral
Bowers et al. (2009) [70]	Case study, 1	The accessory long flexor muscle of the fingers (FDAL) is reported as the cause of tarsal tunnel syndrome and was found running deep to the flexor retinaculum in the form of a muscle belly, posterior to the tendon of the long flexor hallux muscle.	Yes, tarsal tunnel syndrome	Not included	USA	Male	Right
Canter D and Siesel K, (1997) [71]	Cadaveric study, 26	Two cadavers presented unilateral (right) FLAD with different origins and similar insertions. Both entered the tarsal tunnel in the form of a tendon.	Not specified	FLAD was present in 2 of 26 corpses (7.6% of corpses and 3.8% of legs)	USA	Both	Right
Cheung Y et al. (1999) [72]	Descriptive study, 112	Twenty specimens of FDAL muscles were observed. Six cases were collected from a group of 100 asymptomatic patients, and fourteen cases were collected from a group of 12 symptomatic patients presenting with ankle discomfort, observing 2 bilateral cases and 10 unilateral cases.	Yes, tarsal tunnel syndrome	The FDAL muscle prevalence calculated from the group of asymptomatic patients was 6% (6/100).	USA	76 male, 36 female	2 bilateral, 16 unilateral
Deroy A, et al. (2002) [73]	Case study, 1	A cadaveric specimen presented a FLAD muscle that coursed with a single tendon through the tarsal tunnel and then divided into two portions at Henry’s knot.	Not specified	Not included	USA	Not specified	Left
Dobbs M, et al. (2005) [74]	Retrospective review, 835	A FLAD muscle was found in 6.6% (55) of patients who underwent surgery to correct equinus foot. Patients with FLAD were 36% bilateral and 64% unilateral.	Not specified	The comparison group was reflected with an odds ratio of 1.0. For continuous variables, *p* values were based on unpaired *t* tests. *p*, 0.01 was considered statistically significant.	USA	294 female, 541 male	20 bilateral (36%) 35 unilateral (64%)
Duran-Stanton A, et al. (2010) [75]	Case study, 1	The patient suffered from tarsal tunnel syndrome due to the presence of two anomalous accessory muscles deep to the flexor retinaculum: the accessory flexor digitorum longus and the internal peroneocalcaneus.	Yes, tarsal tunnel syndrome	Not included	USA	Male	Left
Hwang S and Hill R, (2009) [76]	Case report, 1	Cadaveric dissection revealed a bilateral FLAD muscle with two heads, entering the tarsal tunnel as a single tendon.	Not specified	Not included	USA	Female	Bilateral
Kubota H (2001) [77]	Case report, 1	An infant presented with clubfoot and Nager syndrome. Corrective surgery revealed the presence of a single-belly FLAD muscle. It is consistent with previous reports that correlate clubfoot with the presence of the muscle.	Not specified	Not included	Japan	Male	Right
Nathan H et al. (1975) [78]	Cadaveric study, 100 (200 lower limbs)	An FDAL muscle was found in 12 cadavers, 10 unilateral and 2 bilateral, for a total of 14 aberrations from 200 dissected legs.	Not specified	The abnormality was found in 12% of cadavers and 7% of dissected legs.	Israel	Not specified	10 unilateral, 2 bilateral
Ogut T and Ayhan E. (2010) [79]	Case report, 1	A FLAD muscle caused tenosynovitis of the flexor hallux longus muscle. Surgical removal of the aberration eliminated symptoms for at least 1 year post-op.	Not specified	Not included	Turkey	Male	Left
Shaheen S et al. (2015) [80]	Prospective study, 261	FLAD was present in 18.39% (48) of patients with resistant clubfoot and 13.14% (411) of legs. The presence of FLAD can be predicted by the Samir-Adams sign.	Not specified	Not included	Sudan	37 male, 11 female	15 right, 27 left, 7 bilateral
Wittmayer B et al. (2007) [81]	Case report, 1	A patient with tarsal tunnel syndrome presented a FLAD muscle with visible entrapment of the posterior tibial nerve by its tendon, proximal to the porta pedis. Tendon transection eliminated preoperative symptoms.	Yes, tarsal tunnel syndrome	Not included	USA	Female	Left
Batista J et al. (2015) [82]	Case report, 1	A FLAD muscle was present in a patient with a history of pain in the posterior region of the ankle and reduced mobility in the fingers. Resection of the flexor retinaculum and posterior talar process alleviated the symptoms.	Yes, posterior tibial neurovascular bundle	Not included	Argentina, Spain	Male	Left
Ettehadi H (2022) [83]	Case report, 1	Surgery to relieve the tarsal tunnel revealed the belly of FLAD involving the posterior tibial nerve. The release of the muscle–nerve junction and neurolysis alleviated symptoms.	Yes, tarsal tunnel syndrome	Not included	South Africa	Female	Right
Holzmann M (2009) [84]	Cadaveric study, 1	Pedagogical dissection revealed a unilateral FLAD of two bellies located under the flexor retinaculum.	Not specified	Not included	USA	Male	Not specified
Mi- Sun et al. (2014) [85]	Cadaveric study, 80	The FLAD muscle was present in 9 of 80 cadavers and classified into three types. Type I originates in the leg and runs without crossing (Type Ia) or crossing (Type Ib) the posterior tibial neurovascular bundle. Type II originates in the tarsal tunnel. All three types can compress the tibial nerve.	Yes, tarsal tunnel syndrome	The FLAD muscle was observed in 9 of the 80 cadavers (11.3%). Type Ia occurred in 3.8% of cases. Type Ib was present in 2.5% of cases. Type II was found in 6.3% of cases.	Republic of Korea	38 male, 42 female	Bilateral
Buschmann W et al. (1991) [86]	Case study, 7	Four cases of accessory soleus muscle were identified, two of them asymptomatic and the other two with physical manifestations of pain and swelling in the ankle. Two cases of peroneus quartus muscle, both asymptomatic, are reported. A case of FDAL muscle was observed presenting with swelling and pain in the medial area of the ankle.	Not specified	Not included	USA	3 male, 4 female	2 right, 2 left soleus muscles; 1 right, 1 left peroneus quartus; 1 right FDAL
Rosson and Dellon, 2005 [87]	Case study 31	The distribution of the superficial peroneal nerve was analyzed in 35 lower extremities surgically treated for peripheral nerve entrapment.	Yes, superficial peroneal nerve	A total of 57% had the superficial peroneal nerve in the lateral compartment, 26% in the anterior and lateral compartment, and 17% in the anterior compartment.	USA	Not specified	Not specified
Dellon A et al., 2002 [88]	Case study, 65 patients and 29 cadavers	The incidence of anatomical variants related to the peroneus muscle was evaluated among a random selection of 29 (bilateral) cadavers and 65 patients who underwent unilateral peroneal decompression to treat the symptoms of that compression.	Yes, common peroneal nerve	A total of 30% of cadavers and 78.5% of patients had a fibrous band on the inferior surface of the superficial head of the peroneus longus. A total of 43% of cadavers and 20% of patients presented a fibrous band in the superficial area of the deep head of the peroneus longus. The origin of the soleus muscle joined the origin of the peroneus muscle in 9% of cadavers and 6% of patients.	USA	Not specified	Not specified
Park JH et al., 2018 [89]	Cadaveric study, 115 lower limbs	The popliteal region and thigh were evaluated in 115 lower extremities of cadavers, and the incidence of anatomical variations of the distal biceps femoris muscle and its relationship with common peroneal nerve entrapment were analyzed.	Not specified	Not included	Republic of Korea	68 male, 47 female	Not specified

## Data Availability

The data presented in this study are available on request from the corresponding author. The data are not publicly available due to privacy.

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
