# Peer review of "The Association between Anatomical Variants of Musculoskeletal Structures and Nerve Compressions of the Lower Limb: A Systematic Review and Meta-Analysis"

_diagnostics, 2024, doi:10.3390/diagnostics14070695_

Round 1

Reviewer 1 Report

Comments and Suggestions for Authors

1. It is suggested to provide the specific search strategy of at least 1 database in order to objectively demonstrate the search process;

2. Only English articles were included, which may cause selection bias;

3. No publication bias test was performed;

4. For the results of meta-analysis with high heterogeneity, it is suggested to supplement subgroup analysis or meta-regression to find the source of heterogeneity.

Author Response

Response to Reviewer 1

Dear, we appreciate your review and comments, since we are convinced that with the suggested changes our study will improve, below I will detail the response to your proposed comments:

  1. It is suggested to provide the specific search strategy of at least 1 database to order objectively demonstrate the search process.

Response: We have added the Medline search link to improve the reliability of one of the search engines, we appreciate this comment as it will improve our study considerably.

  1. Only English articles were included, which may cause selection bias.

Response: Although we have only found articles in English in the selected search engines, we have included the native language of the majority of the authors (Spanish), but we have not found the high number of studies in our search.

  1. No publication bias test was performed.

Response: Dear reviewer, we have added funnel plot analysis to each prevalence meta-analysis.

  1. For the results of meta-analysis with high heterogeneity, it is suggested to supplement subgroup analysis or meta-regression to find the source of heterogeneity.

Dear reviewer, we have added meta-regression analysis, we appreciate your comments

Sincerely

Research team.

Reviewer 2 Report

Comments and Suggestions for Authors

The authors reported a systematic review and meta-analysis regarding the relationship between lower limb nerve compressions and anatomical variations of musculoskeletal systems.
The article's abstract accurately sums up its content, and the title does the same. The acronym PM have to be expained in the abstract.
The introduction describes what the author hoped to achieve and clearly state the problem being investigated: this review aims to outline the primary anatomical variations and morphofunctional changes of the lower limb that compress nearby nerve systems in the leg and foot, thigh, and gluteal regions. The acronym PM have to be expained in the first citation.
Methods: the author accurately explain how the data was collected. The systematic review and meta-analysis were performed and reported according to the Preferred Reporting Items for Systematic Reviews and Meta-Analyses (PRISMA) statement and has been registered in PROSPERO.
Results:  78 studies that met reported inclusion criteria.
Discussion/Conclusions: Conclusion aresupported by the results.

Author Response

Response to Reviewer 2

Dear, we appreciate your review and comments, since we are convinced that with the suggested changes our study will improve, below I will detail the response to your proposed comments, since all of these were positive, we have not responded to any of them, nor have we applied any changes.

Sincerely

Research team.